# Advances in Multicore Fiber Interferometric Sensors

**DOI:** 10.3390/s23073436

**Published:** 2023-03-24

**Authors:** Yucheng Yao, Zhiyong Zhao, Ming Tang

**Affiliations:** 1Wuhan National Lab for Optoelectronics (WNLO), School of Optics and Electronic Information, Huazhong University of Science and Technology, Wuhan 430074, China; 2Optics Valley Laboratory, Wuhan 430074, China

**Keywords:** optical fiber sensors, multicore fiber (MCF), Fabry–Perot interferometer (FPI), Michelson interferometer (MI), Mach–Zehnder interferometer (MZI), supermode interferometer (SMI)

## Abstract

In this paper, a review of multicore fiber interferometric sensors is given. Due to the specificity of fiber structure, i.e., multiple cores integrated into only one fiber cladding, multicore fiber (MCF) interferometric sensors exhibit many desirable characteristics compared with traditional fiber interferometric sensors based on single-core fibers, such as structural and functional diversity, high integration, space-division multiplexing capacity, etc. Thanks to the unique advantages, e.g., simple fabrication, compact size, and good robustness, MCF interferometric sensors have been developed to measure various physical and chemical parameters such as temperature, strain, curvature, refractive index, vibration, flow, torsion, etc., among which the extraordinary vector-bending sensing has also been extensively studied by making use of the differential responses between different cores of MCFs. In this paper, different types of MCF interferometric sensors and recent developments are comprehensively reviewed. The basic configurations and operating principles are introduced for each interferometric structure, and, eventually, the performances of various MCF interferometric sensors for different applications are compared, including curvature sensing, vibration sensing, temperature sensing, and refractive index sensing.

## 1. Introduction

Over the last three decades, a variety of discrete and distributed optical fiber sensors have been extensively investigated, due to their advantages of compact size, light weight, low attenuation, immunity to electromagnetic interference, and multiplexing capability. Optical fiber sensors can detect a wide range of parameters, such as temperature, strain, curvature, refractive index, vibration, torsion, etc. Owing to their outstanding sensing performance, optical fiber sensors have gained increasing attention from both the academic and industrial communities, and so far, many optical fiber sensors have been widely used in complex real scenarios, such as structural health monitoring [1], intrusion detection [2], seismic monitoring [3], etc.

In recent years, multicore fibers (MCFs) have developed rapidly with the growth in the increased requirements of data transmission capacity and then gradually developed in the field of sensing for their unique multicore structure. In the manufacturing process, MCF is typically fabricated by inserting multiple preform rods into a jacket and then collapsing it by melting it before drawing and coating the fibers [4]. Since it has multiple spatial optical channels in only one fiber cladding, as shown in Figure 1 [5,6,7], MCF provides an excellent platform to fabricate diverse functional lab-on-fiber devices, e.g., the unique structure of MCF is particularly suitable to develop in-fiber integrated functional devices, including multicore fiber gratings, multicore fiber interferometers, and many other spatially integrated hybrid devices. Note that MCF has significant advantages over single-core fibers in terms of integration, compactness, stability, robustness, etc. In addition to the structural benefits, MCF has a unique feature, i.e., the bending sensitivity in off-center cores [8]. As a result, MCF-based sensors possess a remarkable capability that allows the measurement of bending-related parameters, e.g., curvature, vibration, fluid flow, vector bending, etc.

As shown in Figure 2, according to different core-to-core distances, MCFs can be classified into two different categories, i.e., weakly coupled MCFs and strongly coupled MCFs [9]. In the weakly coupled MCF, light transmits separately in different fiber cores with low crosstalk between adjacent cores. By using the independent fiber cores as the interference arms of the interferometer, MCF interferometric sensors composed of weakly coupled MCFs can be employed to develop diverse interferometric sensing systems, including the Fabry–Perot interferometer (FPI), the Michelson interferometer (MI), and the Mach–Zehnder interferometer (MZI), with the advantages of compactness, small size, and robustness. In strongly coupled MCFs, the crosstalk between adjacent cores is intentionally introduced by decreasing the core-to-core distance, resulting in the significant enhancement of light coupling between the cores, where light transmits in the form of supermodes. The supermodes in strongly coupled MCFs can also be used to fabricate fiber interferometers, which have also been used for a wide range of sensing applications. In addition to the crosstalk and core-to-core distance that will impact the interference mechanism, MCF has a diversity of materials and structures, including the core/cladding material, core number, core diameter, arrangement, refractive index profile, etc., which can also impact the performance of MCF interferometric sensors; for example, the reduction in cladding thickness will enhance the sensitivity of an MCF interferometric refractive sensor [10].

In this article, a comprehensive review of recent investigations on MCF interferometric sensors is presented. Section 2 will introduce different MCF interferometer structures that are commonly used in fiber optic sensing, including the Fabry–Perot interferometer (FPI), Michelson interferometer (MI), Mach–Zehnder interferometer (MZI), and supermode interferometers (SMIs), where the configurations, operating principles, and representative studies will be presented. In particular, the advantages brought by MCF will be analyzed. Section 3 will summarize, compare different kinds of MCF interferometric sensors, and discuss future prospects, and in Section 4, a conclusion will be given.

## 2. Multicore Fiber Interferometric Sensors

Fiber interferometric sensors have basically the same operating principle as traditional optical interferometers. By replacing traditional discrete optical components with fiber devices, a fiber interferometer can achieve many benefits, including easy alignment, low insert loss, convenient arrangement, high stability, high coupling efficiency, etc. In fiber interferometer systems, the light travels along different optical paths and combines again at the output side, which generates an interference spectrum. External perturbations that are applied to the interference arms may cause property variations of the interferometer, including fiber length, effective refractive index, mode loss, etc., eventually leading to a change in optical path difference (OPD) or optical intensity. The change in OPD and intensity can be precisely retrieved by measuring the far-field interferogram [11,12], the wavelength shift of the interference spectrum [13,14], the light intensity [15,16], the fast Fourier transform (FFT) of light power’s time response [17,18], etc.

Apparently, compactness, stability, simplicity, and robustness play a decisive role in the performance and applicability of optical fiber sensing systems, and these aspects can be improved by using multicore fibers instead of single-core fibers to construct fiber interferometers. Benefiting from containing multiple individual cores in one fiber, MCFs can support the multiple cavities of the FPI, the multiple interference arms of the MI and MZI, and several supermodes in one fiber cladding, which can greatly simplify the structures of sensors and enhance the robustness and compactness of the sensing systems. On the other hand, since the side cores are usually more sensitive to external disturbances, MCF interferometric sensors have intrinsic advantages in sensing applications that are related to bending and strain, e.g., curvature [13,19], vibration [18,20], twisting angle [7,21], fluid flow [19,22], etc. Particularly, MCF interferometric sensors are able to distinguish the direction of bending [15,23], which has enabled many new sensing applications, e.g., 3D shapes [24]. In addition, MCF interferometric sensors have also been used to measure other external environmental parameters, e.g., temperature [25,26], gas concentration [27], refractive index [28,29], and so on. 

Following the brief introduction in Section 1, Section 2 will summarize different MCF interferometric sensor structures that are commonly used in fiber optic sensing, including their configurations, operating principles, and representative studies, as well as the benefits brought by MCFs.

### 2.1. MCF-Based Fabry–Perot Interferometric Sensors 

As shown in Figure 3, Fabry–Perot interferometers (FPIs) can be classified into intrinsic FPIs and extrinsic FPIs according to the formation of the interference cavity. In the intrinsic FPI, the interference cavity is fabricated inside the optical fiber, while in the extrinsic FPI, a microcavity is fabricated between the cleaved optical fiber end and another external reflection surface. In comparison, the intrinsic FPI is generally more robust but requires more high-cost fabrication methods, such as femtosecond laser processing, chemical etching, external fixation using capillaries, etc.

In the FPI, the reflective optical interference intensity can be expressed as
(1)$$I=I_{1}+I_{2}+2\sqrt{I_{1}I_{2}}\cos\varphi$$
where $I_{1}$ and $I_{2}$ are the intensities of the reflective light from two interfaces, and φ is the phase difference between the two reflective lights, which can be defined as
(2)$$\varphi =\frac{4\pi nL}{\lambda }$$
where λ is the wavelength of the light source, L is the length of the cavity, and n is the refractive index of the medium in the cavity. The free spectrum range (FSR) at wavelength λ is given by
(3)$$FSR=\frac{\lambda^{2}}{2nL}$$

One can see from Equation (3) that bending/strain-induced cavity length L variation and environmental-perturbations-induced refractive index n changes will lead to a shift in the optical interference spectrum, which can be used to retrieve the perturbation of the parameters.

In MCF-based Fabry–Perot interferometric sensors, FPIs are usually discrete and parallel in different cores of a weakly coupled MCF [13,14,30,31,32,33]. Instead of using a single FPI, it is able to achieve vector sensing by measuring several parallel FPIs simultaneously in a single MCF, such as bending direction [13,30], torsion [31], flow direction [32], etc. For example, in 2017, Yang Ouyang et al. proposed a directional bending sensor with the structure of a dual air-cavity FPI, which was fabricated by using MCF and a dual side-hole fiber [13], whose structure is shown in Figure 4a. Figure 4b illustrates the bending sensitivity resulting from bending-induced cavity length variation, where each FPI can be regarded as an independent bending sensor, and the differential responses between the FPIs can be used to retrieve the bending amplitude and direction.When the bending direction is perpendicular to the plane of the air cavities, the light transmits along the neutral plane of bending, therefore the sensor is nearly insensitive in the directions of 90° and 270°, while the sensitivity reaches up to maximum in the bending directions of 0° and 180° when it is parallel to the plane of the air cavities. However, the dual-FPI-based bending sensor cannot achieve omnidirectional bending detection. Instead, in 2006, G.A. Cranch et al. proposed a two-dimensional bending sensor by fabricating three FPIs in a four-core MCF. The bending direction can be retrieved by measuring the differential response between the FPIs, thus enabling omnidirectional bending sensing [30]. In addition, in 2016, Guigen Liu et al. proposed a vector flow sensor by analyzing the differential responses between the reflection spectra of multiple extrinsic FPIs that are fabricated at the cleaved-end faces of a seven-core fiber, and the sensor could detect the liquid flow speed and direction simultaneously [32]. Furthermore, in 2021, Guozhao Wei et al. proposed a needle-shaped sensor by etching three individual FPIs on three single-core fibers and then fixing them in a flexible needle parallelly, and the shape of the needle was retrieved with a reconstruction method, which shows potential in medical puncture applications [24]. In general, by making use of the high sensitivity of FPIs to the cavity length variation, the multiple parallel FPIs that are fabricated in a single MCF allow for bending-direction sensing by analyzing the differential responses between the FPIs, which has great potential in precise shape-sensing applications. Furthermore, the high bending sensitivity of FPIs can also be used for vibration sensing. For example, in 2020, Xiandong Ma et al. proposed a broadband ultrasound sensor [34], where the eccentric cores of MCFs were used to fabricate FPIs, and the reverberating noise was decreased from −1 dB to −28 dB.

In addition to bending sensing, parallel FPIs can also be used to measure tiny torsion angles by utilizing the twist-induced cavity length variation. For example, in 2022, Jing Zhang et al. proposed a torsion sensor where four FPIs were made up of vertically cleaved four-core fibers and an angled mirror, and the sensor was demonstrated to be capable of sensing torsion with a range of ±90° and a resolution of 0.01° [31]. As shown in Figure 5, the cavity length of each FPI will change, respectively, when torsion is applied to the four-core fiber and the spectral characteristics of each FPI can be analyzed utilizing a four-core fan-out unit.

Furthermore, in addition to bending and torsion sensing, MCF Fabry–Perot interferometer (MFPI)-based sensors can also be used for axial displacement sensing by changing the distance between the two reflective surfaces of FPIs. For example, in 2019, Cong Zhang et al. demonstrated a displacement sensor by drilling random tiny holes on the facet of a seven-core fiber with a femtosecond laser, as shown in Figure 6, and then, by measuring the reflection spectra of the cavities between the side cores and the target surface, a micro axial displacement sensing application was demonstrated with a substantial reduction in the dead zone and an extension of the measurement range [33]. Thus, MFPIs are able to offer multi-channel reflection spectra for sensing, which turns out to be particularly helpful in the measurement of transverse deformation and axial displacement. In addition, MFPIs can also be used to sense environmental parameters such as liquid refractive index [14]. By fabricating reference and sensing FPIs in a single MCF, a compact sensor head can be achieved. In general, due to the compact reflective structure, heat resistance, corrosion resistance, and high electromagnetic immunity, MFPI-based sensors can be easily deployed in complex actual environments for a variety of sensing applications.

### 2.2. MCF-Based Michelson Interferometric Sensors

Thanks to the multicore structure of MCF, it provides an excellent platform to construct an in-fiber integrated Michelson interferometer by coupling light into different cores so that the cores act as the arms of the Michelson interferometer. Fresnel reflection light is generated on the end face of the cleaved optical fiber, and the reflected light from different cores of the MCF will give rise to interference. Typical coupling methods of MCF-based Michelson interferometers (MMI) are shown in Figure 7. Instead of using fiber couplers, in order to couple light into different cores of an MCF, the widely used approaches including tapering the fiber at the fusion splice region [10,19,22,35,36] or inserting a segment of multimode fiber (MMF) between the single-mode fiber (SMF) and MCFs [6]. These coupling methods have the advantages of compact structure and long-term stability because of the good spatial consistency of the interference arms. Note that by enhancing the coupling efficiency between fiber cores, e.g., the center core and side cores of the MCF, the light power in each interference arm can be more even and the fringe visibility of the interference spectrum can be improved, which is instrumental in the simplicity of demodulation and the accuracy of measurement. 

In addition to the method of tapering, to optimize the interference spectrum quality of the MMI, coupling configurations based on a fiber ball structure have been demonstrated, which have been fabricated on the fusion surface [37] and the reflecting surface [25]. As shown in Figure 8a, in 2016, Li Duan et al. demonstrated an MMI temperature sensor by tapering the fusion-spliced region between the SMF and the MCF and fabricating a spherical end face by arc-fusion splicing. The first coupling appears when the light transmits through the tapered region and the second coupling occurs at the spherical end face when the light is reflected. The light power of the side cores and the center core can be balanced at the spherical end face and, as a result, a high fringe visibility of 25 dB was achieved, as shown in Figure 8b, which is more than 10 dB higher than the normal MCF-based Michelson interferometers [10,35,36,38].

In the MMI, the phase difference φ of reflected light from two cores is governed by
(4)$$\varphi =\frac{2\pi \cdot OPD}{\lambda }$$
where OPD is given by
(5)$$\mathrm{OPD}=2\left(n_{1}L_{1}-n_{2}L_{2}\right)$$
where $n_{1}$ and $n_{2}$ are the refractive indexes of the two optical fiber arms, while $L_{1}$ and $L_{2}$ are the lengths of different arms. The intensity of the interference spectrum is minimal when
(6)$$\varphi =\left(2m+1\right)\;\pi$$
and the wavelength of the dip is determined by
(7)$$\lambda_{m}=\frac{2\cdot OPD}{2m+1}$$

Thus, the FSR of the Michelson interferometer can be expressed as
(8)$$\mathrm{FSR}=\lambda_{m-1}-\lambda_{m}≅\frac{\lambda_{m}{}^{2}}{OPD}=\frac{4\cdot \mathrm{OPD}}{{\left(2m+1\right)}^{2}}$$

Compared with the FPI, the FSR of a Michelson interferometer is determined by the OPD of the two arms rather than the length of a single interference cavity. Therefore, by precisely controlling the length difference between the two arms, the Michelson interferometer can achieve a large FSR while allowing for long sensing arm length. For the wavelength demodulation-based interrogation system, the Michelson interferometer can achieve a large dynamic sensing range without sacrificing the length of the sensor, which can affect the performance in terms of bending/strain sensitivity and resolution. However, due to the short coherence length of the broadband light source, the OPD between different arms needs to be strictly controlled in a small range in order to generate interference, which increases the fabrication complexity of the Michelson interferometer. Fortunately, due to the almost identical optical path length of each core of MCF, an in-fiber integrated Michelson interferometer using MCF is capable of solving this problem perfectly, which allows for long interference arms and controllable OPD simultaneously. When bending is applied to the MMI, according to the photo-elastic effect, the variation in phase difference Δφ can be described as [11]
(9)$$\Delta \varphi =\frac{2\pi }{\lambda }\left(n\Delta L+\Delta nL\right)=\frac{2\pi L}{\lambda }\left(n+n\frac{dn}{d\varepsilon }\right)\Delta \varepsilon$$
where ΔL and Δn are the bending-induced length difference and refractive index difference between the interference arms, while Δε is the bending-induced local tangential strain difference between the interference cores at the bending point. ($n+n\left(dn/d\varepsilon \right))$ is determined by the material of the fiber core and the wavelength of the transmission light [39], which can be regarded as a constant for the fiber cores of a homogeneous MCF. One can see from Equation (9) that the extension of the interference arm length L can enhance the curvature sensitivity. Thus, the MMI is inherently suitable for bending sensing with long interference arms and controllable optical path difference.

In addition to the advantages mentioned above, note that the bending-induced local tangential strain is dependent on the bending direction and curvature [8], which can be described as
(10)$$\varepsilon_{i}=-\frac{d_{i}}{R}cos\left(\theta_{b}-\theta_{i}\right);\varepsilon_{j}=-\frac{d_{j}}{R}cos\left(\theta_{b}-\theta_{j}\right)$$
(11)$$\Delta \varepsilon =\varepsilon_{i}-\varepsilon_{j}$$
where $\varepsilon_{i}$ and $\varepsilon_{j}$ are the bending-induced local tangential strain in core *i* and core *j* of the MCF. $d_{i}$ and $d_{j}$ are the distances between the two cores and the center of the MCF, respectively. For symmetrical fiber cores in the MCF, $d_{i}$ and $d_{j}$ can be regarded as the same. *R* is the radius of applied fiber bending, and $\theta_{b}$ and $\theta_{i}$ are, respectively, the angle of the bending direction and the angular position of core *i*. As shown in Figure 9, an example is given with the transverse geometric distribution of seven-core fiber. Thus, when the bending direction $\theta_{b}$ is changed under the same bending radius, the strain difference Δε between the interference cores will change. As a result, the optical path difference between the interference arms will change with the bending direction, as well as the wavelength shift of the interference spectrum, i.e., the curvature sensitivity of the MMI has direction dependence, which can be used for vector-bending measurements that can measure the curvature and bending direction.

As shown in Figure 10a, in 2006, Libo Yuan et al. demonstrated a curvature sensor based on dual-core fiber MMI [19], where fiber bending gave rise to the change in the optical path difference of the reflected light in the Michelson interferometer. For dual-core fiber, when the bending direction is perpendicular to the plane of the two cores, the cores will not suffer the tangential strain and exhibit no bending response, so, normally, it can only be used for one-dimensional curvature measurement. To solve this problem, as shown in Figure 10b, by simultaneously measuring two Michelson interferometers based on two dual-core fibers, which are placed with the two planes of fiber cores arranged vertically, a directional flow velocity sensor was demonstrated by Libo Yuan in 2008 [22]. Thanks to the long interference arms and controllable optical path difference, MMIs have been widely developed for bending [11,19,22,35,40,41] and vibration sensing [17,42], with high sensitivity, a large dynamic sensing range, direction distinguishment ability, and low cross-sensitivity.

On the other hand, by utilizing the change in phase difference between the cores of MCF that is caused by dynamic bending, the MMI has also been used for vibration detection [17,42]. As shown in Figure 11, in 2018, Zhiyong Zhao et al. proposed a vibration sensor based on a seven-core fiber Michelson interferometer, where a fan-in coupler was used to couple light from different single-mode fiber pigtails to each core of the MCF, and two cores were selected to construct the in-fiber integrated Michelson interferometer. By monitoring the output power and calculating the fast Fourier transform of the acquired vibration signal, the frequency of variation can be retrieved [17]. Since the Michelson interferometer is not fabricated by splicing or tapering the fibers, it has the outstanding advantages of ultra-compact size and high mechanical strength. In addition, benefiting from the high sensitivity to bending, variation can cause a remarkable phase shift in the MMI, and in 2011, Feng Peng et al. demonstrated an accelerator by measuring the phase shift amplitude of the MMI, which possesses the advantages of small size, light weight, and immunity to electromagnetic interference in comparison with a traditional electrical accelerometer [42]. Furthermore, since both cores can be affected equally by the variation in temperature, the accelerator is intrinsically immune to environmental temperature changes, which enables high-precision acceleration measurement.

In addition to the fiber deformation-related measurement mentioned above, the change in the surrounding environment, i.e., refractive index and temperature, can also lead to OPD variation between the side core and the center core of MCF. The light in the side core can interact with the surroundings, while the center core remains less sensitive to the external environment, which leads to the change in OPD and a shift in the transmission spectrum, so the change in surroundings can be retrieved by measuring the wavelength shift [6,10,36]. In 2011, Ai Zhou et al. proposed a refractive index sensor based on an etched asymmetrical dual-core fiber Michelson interferometer, as shown in Figure 12 [36]. Partial cladding of the MCF was removed by chemical etching to enhance the refractive index sensitivity. The thinning of the fiber cladding leads to leakage of the side core mode and, finally, increases the OPD and enhances the refractive index sensitivity. In addition to the measurement of refractive index, when the ambient temperature is changed, the effective refractive index of the outer core and the center core will change due to the thermal expansion and thermos-optic effect, while the outer cores are more sensitive to the temperature, and the variation in temperature will result in the increase in OPD. Therefore, this feature of the MMI has also been used to develop temperature sensors [25,37,38]. 

### 2.3. MCF-Based Mach–Zehnder Interferometric Sensors

The Mach–Zehnder interferometer (MZI) has been extensively employed in the field of fiber optical sensing since it has the benefits of simple structure, high sensitivity, etc. Thanks to the good spatial consistency and almost the same optical path length between the cores in homogeneous MCFs, the MCF-based Mach–Zehnder interferometer (MMZI) has the advantages of easy fabrication, compact size, good robustness, and long-term stability in comparison with the MZI composed by SMF. As shown in Figure 13, in order to couple light into different cores of an MCF to fabricate the MMZI, the commonly used coupling schemes include tapering at the fusion spliced point or splicing a small segment of MMF between the MCF and SMF. The light is divided into different cores of the MCF at the first fusion region, then travels along different cores individually, and couples back together at the second fusion point and generates interference. Despite resembling the operating principle of the Michelson interferometer, the interference of the MZI is generated by transmitted light instead of reflective light, usually enabling the MZI to have a higher signal-to-noise ratio than the MI. In addition, due to the outstanding advantages of the MMZI, i.e., compact size, simple configuration, easy fabrication, long-term stability, high sensitivity, etc., MMZIs have been widely used for sensing applications of various parameters, which can be classified into bending-related parameters and environmental parameters. In recent years, some novel sensor applications based on the MMZI have been proposed, such as vital signs monitoring [43], microsurgery [44], and distributed vibration sensing [45], showing the prospects of the MMZI in practical application scenarios. Moreover, multicore photonic crystal fibers have also been widely used for MMZIs, and have been developed for measuring temperature [46], strain [47], hydraulic pressure [48], refractive index [49], gas [50], bending [51], etc.

Similar to the MCF-based Michelson interferometer, the long interference arms and controllable optical path difference enable the MMZI to be widely used for bending-sensing applications [12,23,51,52,53,54,55,56,57,58,59,60,61,62,63,64,65,66,67,68,69]. In 2019, Lei Deng et al. proposed a directional bending sensor with the structure of a three-core-fiber-based MZI, as shown in Figure 14a [53]. By laterally offset-splicing a segment of three-core fiber (TCF) between two SMFs, the light can be split into two cores of the TCF from the input SMF and then pass through the TCF before coupling into the output SMF to generate interference. As shown in Figure 14b, with bending applied, the side core and the inner core will be stretched and extruded, respectively, leading to variation in the optical path difference between the two arms of the MMZI [53]. As a result, the optical path difference will lead to a shift in the interference spectrum, and eventually, the curvature can be retrieved from the wavelength shift. By making use of the same principle in Equation (9), where the strain difference Δε between the interference cores is dependent on the bending radius and direction, the wavelength shift of the interference spectrum will change with the variation in bending direction. Thus, the curvature sensitivity of the MMZI will exhibit direction dependence, which allows for directional bending sensing [53,65,66]. Apart from the bending direction discrimination ability, since the symmetrical cores of MCF have the same sensitivity to their surroundings, e.g., ambient temperature, a curvature sensor with low-temperature cross-sensitivity has been demonstrated [52,62,65], which can compensate for the temperature perturbations and enhance the accuracy of curvature measurement. Moreover, it has been demonstrated that tapering the sensing part of MCF can increase the leakage of the high-order mode of each core to enhance the bending sensitivity. In 2021, Yancheng Ji et al. demonstrated an extremely sensitive curvature sensor with a sensitivity of 174.02957 nm/m^−1^, which is three or four orders higher than a normal MMZI. The ultrahigh sensitivity of the curvature sensor was achieved due to the structure of the tapered seven-core fiber MMZI, in which the waist diameter of the sensing part was reduced to about 2 µm [62]. In addition, the high sensitivity of bending sensing also enables the MMZI to be used for vibration detection, which has great potential in practical applications, e.g., the long-term monitoring of construction, mechanical structures, and so on [43,70,71]. As shown in Figure 15, in 2020, Fengze Tan et al. proposed contactless vital signs monitoring, based on few-mode fiber and MCF interferometers [43], where the interferometer is placed under the mattress. Human breathing and heartbeat change slightly the pressure that applies to the interferometer, which eventually causes variation in the output waveform. Because of the inherent frequency difference between respiration and heartbeat, the raw data are filtered to recover respiration and heartbeat signals, respectively.

The unique structure of MCF enables the MMZI to measure torsion, as well [7,21,72,73,74]. For example, as shown in Figure 16a, in 2018, Hailiang Zhang et al. proposed an MMZI-based torsion sensor fabricated by a helical seven-core fiber, in which the side cores are twisted into a spring structure and the center core is almost straight [21]. When external torsion is applied to the fiber, the photo-elastic effect results in a refractive index change in the side cores and cladding, but since the center core is located in the neutral axis of the fiber, theoretically, the torsion-induced refractive index change in the center core is much smaller, and as a result, the change in phase difference between the arms leads to a wavelength shift in the interference spectrum. Due to the photo-elastic effect, the refractive index of the side-core has a positive dependence on the twist rate. Since clockwise and counterclockwise twists give rise to different twist rate variations, i.e., either an increase or decrease in the twist rate, the torsion direction can be determined from the wavelength shift direction of the interference spectrum. As a result, torsion direction discriminative measurement is enabled by using the pre-twisted seven-core fiber MZI sensor. Additionally, since axial strain can also result in variation in the twist rate and the refractive index of the side cores, the helical seven-core-fiber-based MZI can also be used to measure axial transverse strain [75]. Moreover, by making use of the multiple channels in the MCF to generate multiple-path interference, which may have differential responses to external perturbations, e.g., the center core and the side core have different temperature sensitivities, the MCF-based MMZI can achieve multi-parameter measurement. For example, as shown in Figure 16b, since the axial strain sensor is based on a multi-path interference MMZI, which possesses several resonance peaks/dips that have different sensitivities to multi-perturbations, the sensor can also be used to distinguish multiple parameters simultaneously by tracking different resonance peaks/dips in the interference spectrum. For example, an MMZI with multi-path interference was proposed to sense axial/transverse strain and environmental temperature simultaneously [44,47,74,75,76]. Furthermore, as for multipath interference-based MMZI bending sensors, since bending and temperature can both cause different refractive indexes and length variations in each core, sensors that can achieve bending and temperature measurement have been demonstrated [54,55,56,57,64].

Similar to the MCF-based Michelson interferometric sensors, in addition to the curvature/strain-related parameters, such as bending, vibration, axial strain, and torsion, the MMZI is also capable of measuring surrounding environmental parameters, such as refractive index [28,29,77,78,79,80,81,82,83,84,85], ammonia concentration [27], salinity [86], temperature [26,87,88,89,90], magnetic field [91], etc. Because the side cores are usually more sensitive to external disturbances, while the center core remains less sensitive to the external environment, which leads to the change in OPD between the cores and a shift in the transmission spectrum when an environmental parameter changes, the change in external parameters can be retrieved by measuring the wavelength shift of the interference spectrum. On the other hand, etching the cladding or tapering at the sensing region of the MCF can reduce the thickness of the fiber cladding and cause mode leakage of the side cores, which has been used for sensitivity-enhanced refractive index sensing [29,77,78,79,80,81,82,83,84,85] and temperature sensing [26,87,88]. Furthermore, by using special materials to coat the etched or tapered MCF, the sensing sensitivity of targeted parameters can be further improved [26,27,82,91]. For example, the graphene coating can be regarded as a high-index monolayer, which can enhance the evanescent field and increase the interaction between the side cores and the ambient refractive index. In 2020, Donglai Guo et al. proposed a refractive index sensor fabricated by coating the tapered MCF with graphene, as shown in Figure 17, and the result verifies that the evanescent field could be enhanced and the refractive index sensitivity could be improved [82]. In addition to graphene, magnetic fluid is another material that has been widely used to develop MCF sensors. Magnetic fluid is a kind of colloidal solution with magnetic nanoparticles that can aggregate into clusters directly under the applied magnetic field and change the ambient refractive index. In 2018, Chunxia Yue et al. proposed a magnetic sensor fabricated by immersing a segment of tapered MCF into magnetic fluid, as shown in Figure 17b. By measuring the refractive index variation of the magnetic fluid with a tapered MMZI instead of the conventional method of measuring the Faraday rotation of the light, a high-sensitivity MMZI-based magnetic field sensor was achieved [91].

In addition to the point sensors, the MCF-based Mach–Zehnder interferometer can also be used for distributed sensing. Due to the spatial length consistency of cores in the MCF, it can be used to implement an MZI that has very long-distance arms. For example, in 2021, Zhiyong Zhao et al. proposed a novel long-range distributed dual MZI vibration sensor, which consisted of two counter-propagating interferometers that were space-division multiplexed in different cores of a weakly coupled seven-core fiber, as shown in Figure 18 [45]. The dual MZI vibration sensor can determine the vibration location through the time-delay difference of two vibration signals in the two MZIs by calculating the cross-correlation of the acquired data, and no additional complicated data processing is required. Since forward-transmitting CW light rather than weak backscattering light is used, together with the advantages of simple structure, a high signal-to-noise ratio, and high sensitivity, the proposed MCF-based dual MZI vibration sensor shows great potential in the application of ultra-long-distance distributed sensing.

### 2.4. Supermode Interferometer-Based (SMI) Sensors

Light transmits independently in different cores of a traditional weakly coupled multicore fiber (MCF), with low crosstalk between cores due to large core-to-core distance. However, in recent years, a new type of strongly coupled multicore fiber with intentionally closely arranged cores has also been proposed, which supports a special mode guiding status, i.e., supermode [92]. The supermode usually has a large effective area because it can be regarded as a superposition of isolated modes of each MCF core. As an example, Figure 19 shows the simulated optical field distribution of supermodes that are excited in three-core fiber and seven-core fiber, respectively [93,94]. The strongly coupled multicore fiber is considered a form of multimode fiber, in which the supermodes can be used to develop modal interferometers for sensing applications. Typical supermode interferometric sensors include transmissive and reflective structures, as shown in Figure 20, i.e., SMF–MCF–SMF (SMS) transmissive structure and SMF–MCF reflective (SMR) structure. The difference in the propagation constants of the supermodes will cause a phase difference in transmission, so the MCF acts in a similar manner to a directional coupler [95,96] and the interference spectrum of the SMI exhibits a periodic maximum and minimum. The changes in fiber length, internal stress, and the effective refractive index of the supermodes will alter the interference spectrum of the interferometer and the coupling power of the supermodes, which can be used to retrieve the sensing parameters. Since different supermodes in the MCF have distinct propagation constants and different sensitivities to external environmental parameters, the external environmental variation can also result in a phase difference between the supermodes, which can be used for the detection of surrounding environmental parameters such as temperature [97,98,99,100,101,102,103,104], refractive index [105], and, simultaneously, temperature and refractive index sensing [106].

Thanks to the large spatial distribution of the fiber mode field of supermodes, bending will easily distort the mode field, which can change the mode loss of each supermode and the phase difference between the supermodes. As a result, bending will result in wavelength shifts [93], light power variation [15], and fringe visibility changes [107] in the SMI, which has been widely used for the high-sensitivity measurement of bending/strain-related external perturbations, such as bending [15,93,94,95,96,107,108,109,110,111,112], vibration [16,18,20,113,114,115,116,117], and strain [118,119]. Furthermore, by using an MCF with asymmetrically distributed cores, the SMI can be used to sense the bending direction. For example, in 2020, Josu Amorebieta et al. proposed an SMI-based vibration sensor, where an asymmetric three-core strongly coupled multicore fiber was used to fabricate a reflective supermode interferometer [20], and this work indicates that the wavelength shift direction of an asymmetric SMI is dependent on bending direction, as shown in Figure 21. When bending is applied in different directions, the effective refractive index and the mode loss of each supermode will change, resulting in the wavelength shift and the coupled light power being direction dependent. For this reason, asymmetrical strongly coupled MCF can be used for vector-bending sensors that can detect the curvature and distinguish the bending direction simultaneously [15,93,111]. In addition to using the MCF with an asymmetrical core arrangement, symmetrical seven-core fiber can also be used for vector-bending sensing by using a femtosecond laser to modify the refractive index of one side core. As a result, an asymmetrical supermode can be generated, which is composed of the modal of six side cores, and the bending direction can be distinguished [112]. Furthermore, the axial stress can also change the refractive index of the fiber cores and increase the phase difference between the supermode, so SMIs can also be used for measuring axial strain [115,118] and force [119]. 

In addition to the bending and strain sensing applications, supermode interferometers are also capable of sensing the bending-related dynamic parameters, i.e., vibration [16,18,20,113,114,115,116,117]. Due to the large mode spatial distribution of SMIs, bending can easily cause variation in the interference spectrum and the output optical power. Thus, by monitoring the wavelength shift through a high-speed spectrometer [18,20,113,114,115,116,117] or detecting the output optical power variation [16,20], the vibration frequency can be retrieved through the fast Fourier transform (FFT) of the time domain signal. For example, in 2017, Joel Villatoro et al. proposed a cantilever vibration sensor fabricated by a strongly coupled MCF, as shown in Figure 22a [115]. When vibration is applied to the supermode interferometer, the interference spectrum will undergo a periodic reciprocating shift, as shown in Figure 22b, and the vibration frequency can be obtained by calculating the FFT of the acquired vibration waveform. Because the mass of the cantilever will decide the resonance frequency of the SMI, which can be easily adjusted by changing the length of MCF, the SMI-based vibration has a large tunable frequency measurement range from the order of mHz [20] to KHz [18], theoretically. To further improve the bending sensitivity of the SMI, a phase-shifted modal interferometer has been proposed, which is composed of two separated SMIs in series [16,20,115]. Due to the extended cantilever, the modal phase-shift interferometer has higher sensitivity.

## 3. Discussion and Comparison

Thanks to the multicore structure of MCF, it offers an excellent platform to develop diverse functional fiber devices, e.g., Fabry–Perot interferometers, Michelson interferometers, Mach–Zehnder interferometers, and supermode interferometers, which have much better flexibility and diversity than conventional single-mode fiber. By space-division multiplexing in multicore fibers, several individual interferometers can be constructed in a single fiber cladding, which greatly improves the compactness and integration of the sensor head. Furthermore, the differential responses in fiber cores can provide more detailed information for measurement, which can improve the performance of the sensor, e.g., compensation for environmental perturbations. Furthermore, the outstanding spatial consistency of MCF provides a controllable optical path difference between the interference arms, which greatly simplifies the fabrication of Michelson interferometers and Mach–Zehnder interferometers. Thus, in comparison with the interferometers composed of single-mode fiber, MCF interferometers have the advantages of easy fabrication, compact size, light weight, high robustness, stability, etc. 

Due to the differential responses to external perturbations between the cores of MCF, especially for the center core and the side cores, MCF interferometric sensors have significant advantages in two types of sensing applications, i.e., monitoring environmental changes, including temperature, refractive index, etc., as well as measuring bending and related parameters, including curvature, vibration, strain, etc. Particularly, MCF interferometric sensors are inherently suitable for vector-bending measurement, which greatly widens the applications for curvature sensing, such as 3D shape sensing. Nowadays, with the growing demand for large-scale sensing, the applications of MCF interferometric sensors are no longer limited to point sensing. Long-distance distributed sensing has also been demonstrated using two counter-propagating MCF Mach–Zehnder interferometers [93]. This opens a new way to achieve an ultra-long-distance distributed sensor that overcomes the restriction of sensing range imposed by the weak backscattering signals in the existing OTDR-based distributed sensing schemes. 

The reported four different structures of MCF interferometric sensors exhibit their own advantages and disadvantages. For example, MCF-based FPIs are able to achieve a very tiny size of the sensor head (less than 200 μm [13,14,24,31,32,34]), and each core can be used to fabricate an individual sensor, which can measure differential responses in distinct cores separately [13,14,24,31,32,33]. However, the manipulation of an individual core in the MCF is still a technical challenge, thus it is difficult to fabricate intrinsic FPIs in different cores of MCF. In comparison with MFPIs, MCF-based Michelson interferometers and Mach–Zehnder interferometers have the prominent advantage of easy fabrication. Furthermore, the usage of MCF fan-in/fan-out couplers makes it easy to select specific cores in the MCF to construct MMIs [17] and MMZIs [45], and the stability can be improved since there is no need to taper the fiber. Thanks to the forward-transmitting structure, MMZIs usually have a higher signal-to-noise ratio than MMIs. However, as for bending/vibration measurements, since MMIs and MMZIs require at least two arms to generate interference, the angular deflection of the original fiber placement, which is difficult to calibrate, will impact the accuracy of the measurement. On the other hand, the reported supermode interferometers normally have a very simple structure. However, the accuracy of SMI sensors is restricted by cross-sensitivity due to the different sensitivities to the external environment of different supermodes in the MCF, e.g., the center-core supermode and the symmetrical side-core supermode in the seven-core fiber, as shown in Figure 19b. In comparison, the typical cross-sensitivity between temperature and bending can be easily compensated for in MMIs and MMZIs by selecting symmetrical cores as interference arms, and in MFPIs, the environmental parameters can be compensated for by using the differential responses of two individual FPIs in symmetrical cores. It should be noted that the interferometer is polarization-sensitive and the polarization state needs to be adjusted during the experiment in order to obtain high fringe visibility.

In order to evaluate the performance of various multicore fiber interferometric sensors, comparisons have been carried out according to several specific application fields, including curvature, vibration, temperature, and refractive index sensing, which are listed in Table 1, Table 2, Table 3 and Table 4. MCF interferometric sensors are compared in terms of sensor structure design, sensitivity, applications, etc.

In MCF-based interferometers, due to the structural characteristics, external perturbations could easily give rise to variations in the optical phase difference between the arms; therefore, MCF-based curvature sensors have been extensively studied, exhibiting the advantages of compact size, high sensitivity, and direction dependence. Table 1 summarizes the MCF interferometer-based curvature sensors. Most sensors have a sensitivity of more than 20 nm/m^−1^, and the highest bending sensitivity is about 174.03 nm/m^−1^ [62], which is achieved by tapering the MCF to increase the leakage of the high-order mode and thus increase the bending sensitivity. Note that as in the actual applications, bending may occur in various and unexpected directions, so vector-bending sensors that can detect the bending direction and curvature simultaneously are of great significance. Fortunately, MCFs are inherently suitable for fabricating vector-bending sensors with simple configurations and easy fabrication because the inherent strain in each core is decided by the bending direction and curvature simultaneously. However, despite the advantages of MCF interferometric bending sensors mentioned above, it is noteworthy that the original fiber placement will influence the accuracy of bending sensing. Unfortunately, due to the small size of the fiber, high-precision angular position calibration is difficult to achieve, which may cause errors in bending measurements. In addition, the tiny twist of MMZIs and MIs may also cause errors in bending measurements.

Based on the bending sensitivity of MCF interferometers, MCF-based interferometric sensors can also be used to measure vibration. Since vibration will lead to the deformation of the sensing fiber with fast bending changes, the interference spectrum will undergo a periodic reciprocating shift with bending, and by calculating the FFT of the acquired vibration waveform, the vibration frequency can be obtained. In addition to the periodicity of vibration, the amplitude of vibration, which indicates the magnitude of the restoring force, can also be used for acceleration measurement [20]. It has been widely reported that high-sensitivity vibration measurement can be achieved by using MCF interferometer-based sensors [16,17,20,43,45,70,71,113,114,116,117] because these sensors are sensitive to curvature changes, which will contribute to the high sensitivity of the vibration sensors. Table 2 summarizes various MCF interferometer-based vibration sensors. It indicates that an MCF-based interferometric vibration sensor can provide a large frequency-measurement range, varying from 1 mHz [20] to 12 kHz [17]. These compact and high-sensitivity sensors can be used in many fields, such as the long-term monitoring of construction and mechanical structures [71], contactless vital signs monitoring [43], seismic measurement [116], accelerometers [20], etc. 

Due to the structural characteristics of MCF, the side cores are usually more sensitive to environmental disturbances, while the center core remains less sensitive to the external environmental due to its longer distance from the outer cladding. Thus, MCF-based interferometers can be used to develop temperature sensors. Table 3 summarizes various MCF interferometer-based temperature sensors, where the temperature sensitivity varies from 29.426 pm/°C [100] to 25 nm/°C [26]. It indicates that etching and tapering the MCF are two effective methods to improve the temperature sensitivity, as they can enhance the interaction between the side cores and the surrounding environment, thus resulting in higher temperature sensitivity. In addition, by covering the tapered MCF with a flexible thermo-optical material, the temperature sensitivity can be further improved [26].

Similar to the same principle of MCF-based interferometric temperature sensors, variation in the surrounding refractive index can also increase the phase difference between the side cores and the center core of the MCF, which can be used to measure the refractive index. Table 4 summarizes different kinds of MCF interferometer-based refractive index sensors, where the reported refractive index sensitivity varies from 113.27 nm/RIU [58] to 35,089.28 nm/RIU [80]. Since the light in the outer core will be influenced by the environmental refractive index and the reduction in cladding thickness will enhance the impact between the outer core and the surroundings, cladding etching and tapering of the MCF are widely used to enhance the sensitivity to the surrounding refractive index. For example, by tapering a trench-assisted MCF with a waist diameter of 25 μm and a length of 15 mm, the refractive index sensitivity was improved to 35089.28 nm/RIU [80]. Moreover, by covering the tapered MCF with graphene coating, the evanescent field can also be enhanced to increase the interaction with the environment, which can achieve a sensitivity of 9194.6 nm/RIU [82]. However, despite tapering and etching the cladding being helpful ways to enhance sensitivity for both MCF interferometric refractive index sensors and temperature sensors, the mechanical strength of the sensor head is sacrificed, which is not conducive to long-term measurements in complex environments.

## 4. Conclusions

In conclusion, the advances in multicore fiber interferometric sensors are summarized thoroughly in this paper. Thanks to the multicore structure of MCF, it has been used to develop diverse in-fiber integrated functional fiber devices, including Fabry–Perot interferometers, Michelson interferometers, Mach–Zehnder interferometers, and supermode interferometers, which have been widely used for sensing applications with some attractive advantages, e.g., compact size, light weight, easy fabrication, low cost, high sensitivity, and, particularly, allowing for vector-bending sensing. MCF interferometric sensors are capable of measuring a wide range of parameters, including temperature, strain, curvature, refractive index, vibration, torsion, etc. It is believed that multicore fiber interferometric sensors will find more applications in real scenarios in the future.

## Figures and Tables

**Figure 1 sensors-23-03436-f001:**
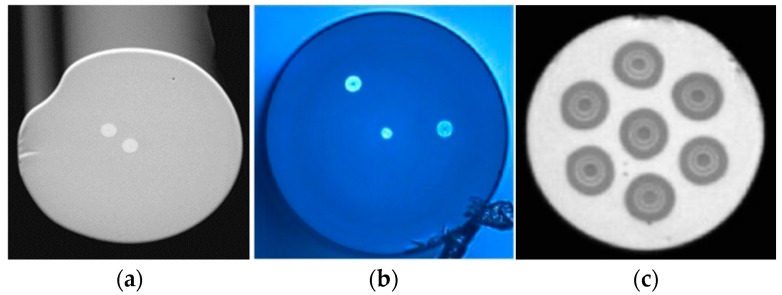
Cross-sectional view of several MCFs. (**a**) Asymmetrical dual-core fiber [5], (**b**) V-type three-core fiber [6], (**c**) weakly coupled seven-core fiber [7].

**Figure 2 sensors-23-03436-f002:**
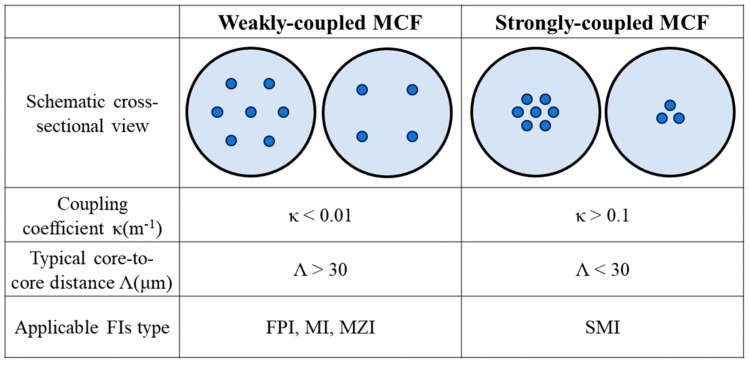
Classification of MCF for interferometers [9].

**Figure 3 sensors-23-03436-f003:**
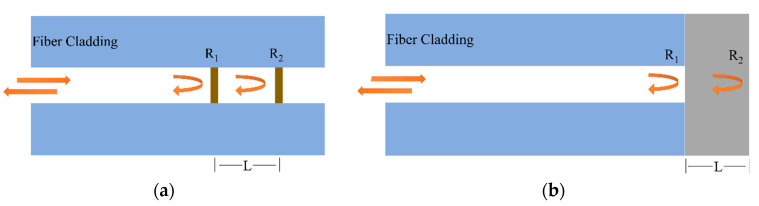
Schematic diagram of (**a**) an intrinsic FPI and (**b**) an extrinsic FPI.

**Figure 4 sensors-23-03436-f004:**
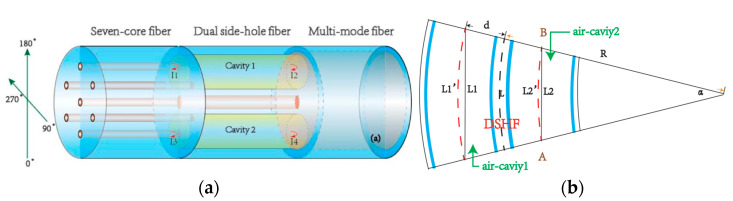
(**a**) Schematic of the dual air-cavity FPI and (**b**) a schematic diagram of bending [13].

**Figure 5 sensors-23-03436-f005:**
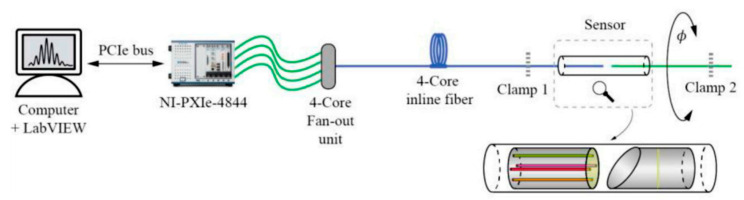
Schematic of the torsion sensor based on FPIs made up of a 4-core fiber [31].

**Figure 6 sensors-23-03436-f006:**
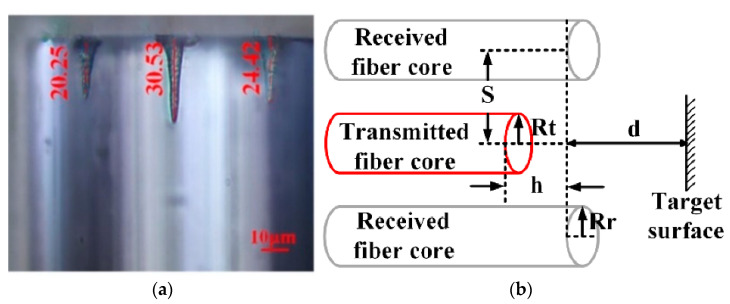
(**a**) Side view of the seven-core fiber after individual micro-hole fabrication; (**b**) schematic design of the axial displacement sensor head [33].

**Figure 7 sensors-23-03436-f007:**
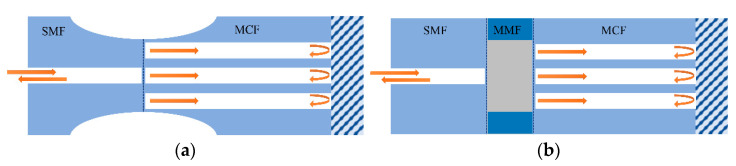
Schematics of typical coupling methods of the MCF-based Michelson interferometer. (**a**) Tapering the MCFs at the fusing point, (**b**) fusing a segment of MMF between the SMF and MCFs.

**Figure 8 sensors-23-03436-f008:**
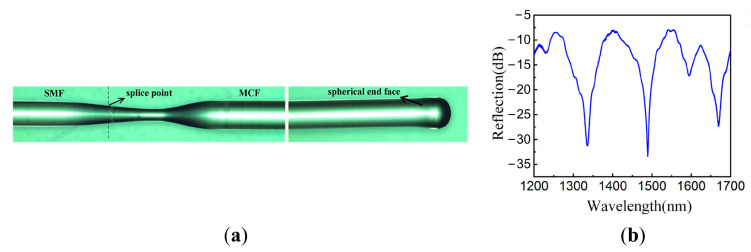
(**a**) Schematic of the high-temperature sensor head utilizing a seven-core fiber Michelson interferometer. (**b**) The typical reflectance spectrum of the seven-core fiber Michelson interferometer with a spherical end face [25].

**Figure 9 sensors-23-03436-f009:**
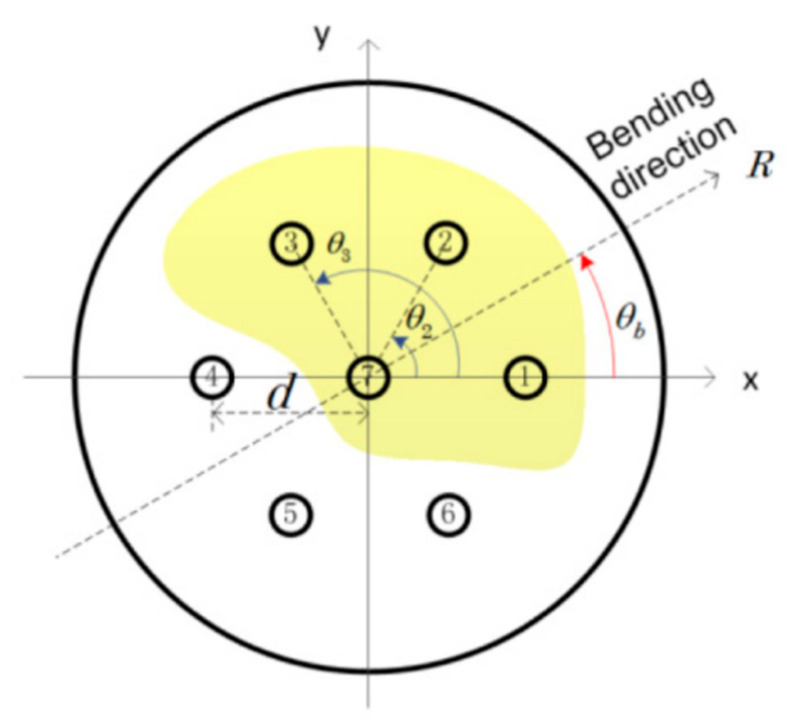
Transversal distribution of the fiber cores with the definition of the important geometrical parameters in seven-core fiber [8].

**Figure 10 sensors-23-03436-f010:**
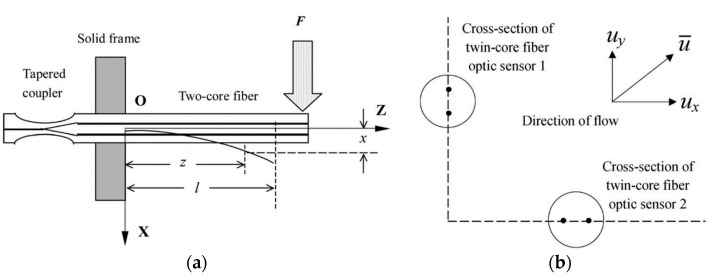
(**a**) Experimental setup of the dual-core-fiber-based Michelson interferometer bending sensor. The inset shows the bent dual-core fiber [19]. (**b**) The schematics of the arrangement of the two dual-core fibers for a vector flow velocity sensor [22].

**Figure 11 sensors-23-03436-f011:**
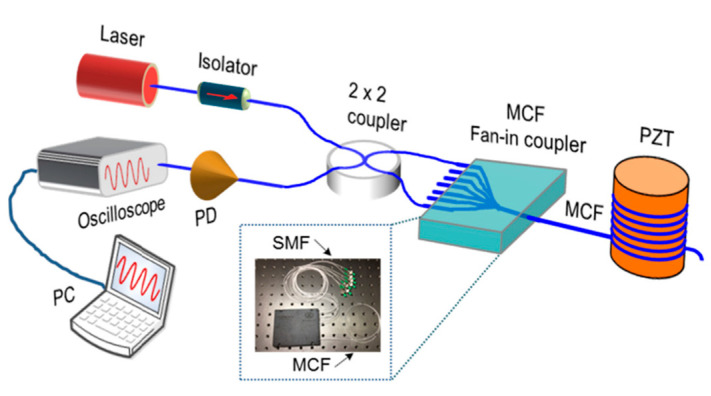
Experimental setup for the MMI vibration sensor [17]. PD: photodetector; PC: personal computer. The inset shows the packaged MCF fan-in coupler.

**Figure 12 sensors-23-03436-f012:**
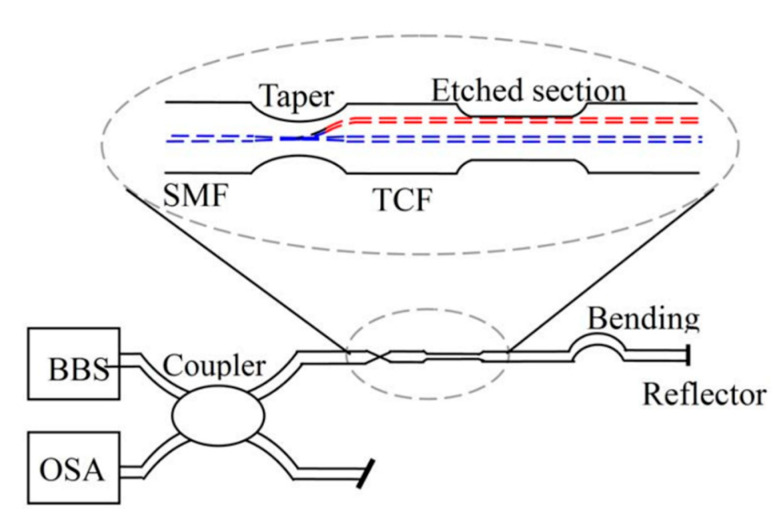
Experimental setup for the etched asymmetrical dual-core fiber Michelson interferometer refractive index sensor [36].

**Figure 13 sensors-23-03436-f013:**
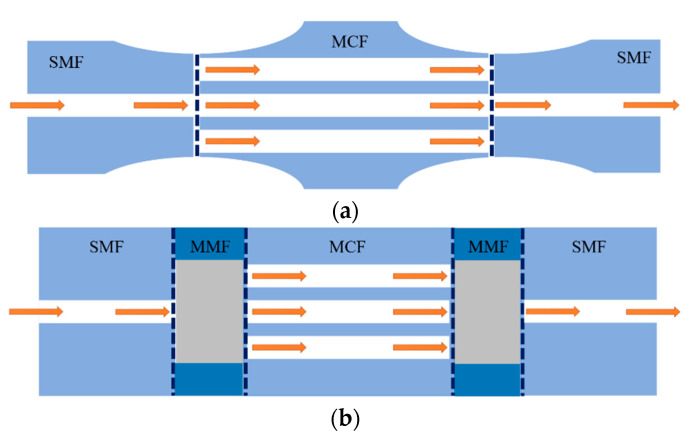
Schematics of the typical coupling methods of the MCF-based Mach–Zehnder interferometer. (**a**) Tapering the MCF at the fusing point, (**b**) fusing a segment of MMF between the SMF and MCF.

**Figure 14 sensors-23-03436-f014:**
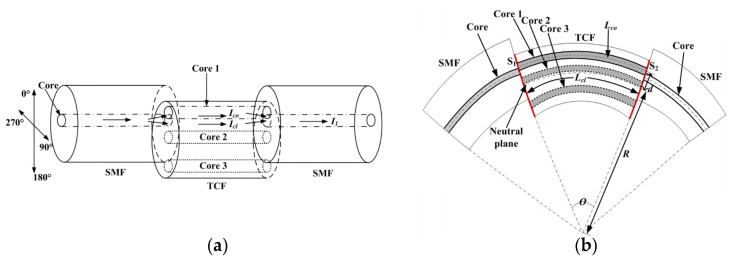
(**a**) Schematics of a three-core-fiber-based MZI for directional bending sensing. (**b**) An illustration of the bent three-core MZI [53].

**Figure 15 sensors-23-03436-f015:**
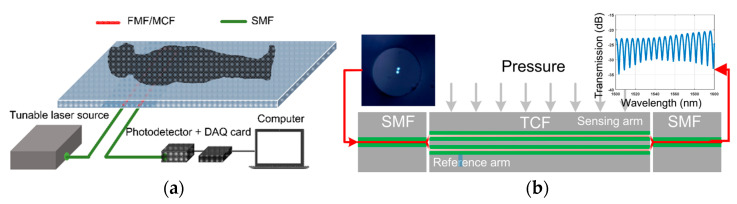
(**a**) Illustration of the vital signs monitoring experiment. The twin-core fiber (TCF)-based MZI is placed under the mattress. DAQ: data acquisition. (**b**) Schematic diagram of twin-core fiber MZI vibration sensor; the inset shows the interference spectrum of the measured twin-core-fiber-based MZI [43].

**Figure 16 sensors-23-03436-f016:**
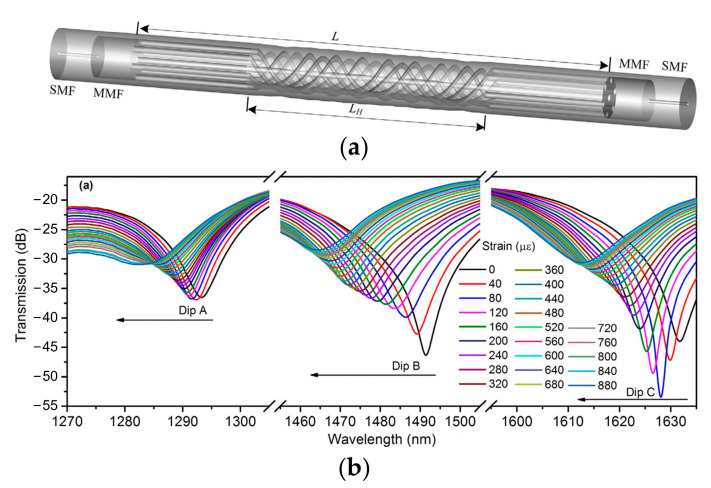
(**a**) The configuration of the proposed torsion sensor and axial strain sensor based on SCF with a helical structure [21,75]. (**b**) The axial-strain-induced wavelength shifts of dip A, dip B, and dip C with different sensitivities [75].

**Figure 17 sensors-23-03436-f017:**
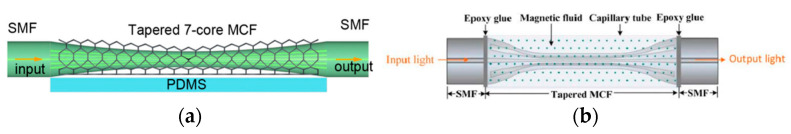
(**a**) Schematics of the tapered-MMZI-based refractive index sensor, in which the tapered part of the MCF is graphene-coated [82]. PDMS: polydimethylsiloxane substrate. (**b**) Schematic of the tapered-MMZI-based magnetic-field sensor, in which the sensing part is immersed in epoxy glue filled with the magnetic fluid [91].

**Figure 18 sensors-23-03436-f018:**
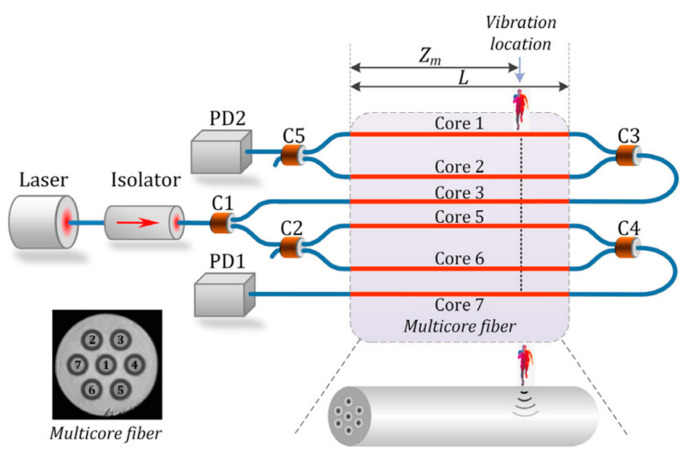
Schematic diagram of the proposed long-range distributed vibration sensor. The inset shows the cross-sectional view of the MCF used. C1–C5: optical couplers [45].

**Figure 19 sensors-23-03436-f019:**
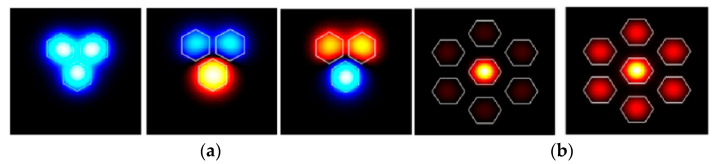
Simulated supermodes in the strongly coupled MCF. (**a**) The orthogonal supermodes in the three-core fiber [93], and (**b**) the center-core supermode and the symmetrical side-core supermode in the seven-core fiber [94].

**Figure 20 sensors-23-03436-f020:**
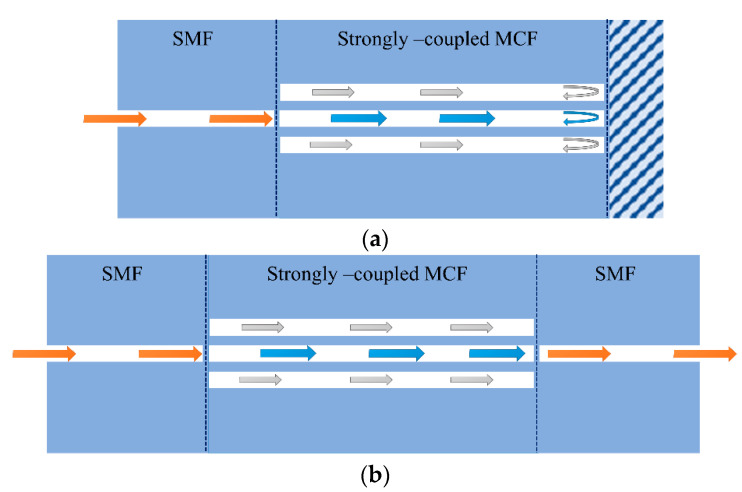
Schematics of a supermode interferometer with (**a**) a reflected structure; (**b**) a transmitted structure.

**Figure 21 sensors-23-03436-f021:**
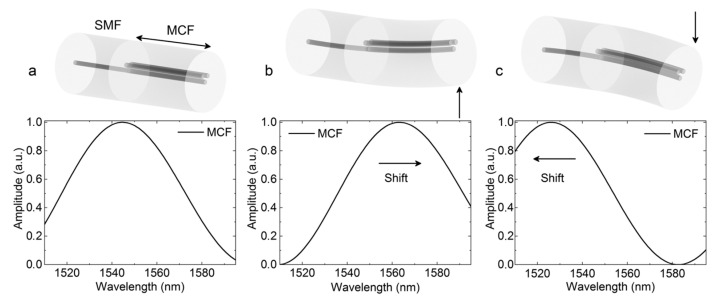
Simulated optical spectrum of the three-core-fiber-based supermode interferometer when the MCF is (**a**) straight, (**b**) bent upwards, and (**c**) bent downwards [20].

**Figure 22 sensors-23-03436-f022:**
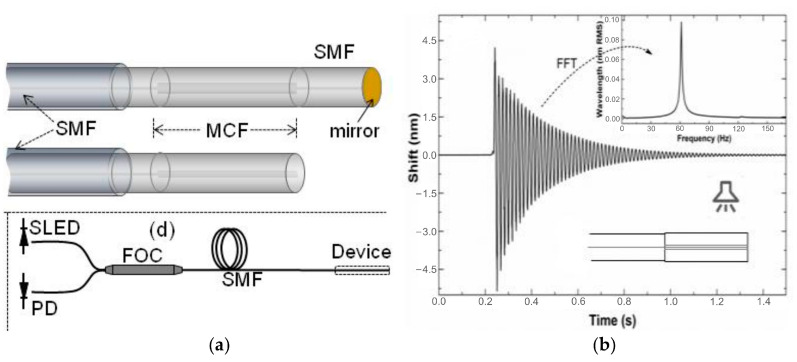
(**a**) Illustration of the vibration sensor heads, including the SMF–MCF–SMF structure and the SMF–MCF structure, and the schematic of the experimental setup. (**b**) The wavelength shift vs. time of the SMI when it was vibrating freely, and the inset shows the FFT of the time domain signal [115].

**Table 1 sensors-23-03436-t001:** Comparison of MCF interferometer-based curvature sensors.

Sensor Structure	Sensitivity (nm/m^−1^)	Applications	REF.
Four-core MCF MI	―	Omnidirectional curvature sensing	[11]
Dual air-cavity FPI *	242.5 pm/m^−1^	Simultaneous measurement of strain and curvature	[13]
Asymmetrical three-core fiber SMI	507 pm/° and 587 nW/°	Bending angle and direction sensing	[15]
Three parallel FPIs	― *	3D shape sensing in mini-invasive surgery	[24]
Tapered dual-core fiber MI *	―	Flow velocity sensing	[22]
Tapered three-core fiber MI	−95.22 pm/m^−1^	Curvature sensing	[35]
Plastic dual-core fiber MI	―	Nano-displacement/curvature sensing	[40]
Three-core PCF MI	―	Omnidirectional curvature sensing	[41]
MMF-coupled three-core fiber MZI *	−28.29	Low-temperature crosstalk curvature measurement	[52]
Lateral offset-spliced three-core fiber MZI	−20.48	Biomedical sensing equipment	[53]
MMF-coupled seven-core fiber MZI	31.54	Curvature measurement sensing	[56]
MMF-coupled seven-core fiber MZI	41.46	Temperature and curvature sensing	[57]
Four-core fiber MZI	2.55	Strain, refractive index, and curvature	[58]
Lateral offset-spliced seven-core fiber MZI	25.96	Strain and curvature sensing	[60]
Twisted seven-core fiber MZI	−25.16	Structural health monitoring	[61]
Tapered seven-core fiber MZI	174.03 nm/m^−1^	Low-temperature crosstalk bending measurement	[62]
Dual-core fiber MZI	1.52	Large curvature measurement	[63]
Hump-shaped dual-core fiber MZI	−6.18	Directional curvature measurement	[65]
Hole-assistant dual-core fiber MZI	−15.95	Low-temperature/refractive index crosstalk vector-bending sensor	[66]
Asymmetrical dual-core fiber MZI	12.98	Directional curvature measurement	[68]
Asymmetrical three-core fiber SMI *	4094 pm/° and 1.99 nm/m^−1^	Bending angle and orientation sensing	[93]
Asymmetrical dual-core fiber SMI	−137.88	Small curvature with high sensitivity	[95]
Seven-core fiber SMI	3000 nm/mm^−1^	Structural monitoring	[96]
Two asymmetrical three-core fiber SMIs	4.66 dB/m^−1^	Directional bending measurement and vibrations, force, pressure, etc.	[111]
Femtosecond laser-writing-modified seven-core fiber SMI	1.4 nm/° and 17.5 nm/m^−1^	Omnidirectional curvature and small bending angle measurement	[112]

* “―” means “not mentioned”; Fabry–Perot interferometer (FPI); Michelson interferometer (MI); Mach–Zehnder interferometer (MZI); supermode interferometer (SMI).

**Table 2 sensors-23-03436-t002:** Comparison of MCF interferometer-based vibration sensors.

Sensor Structure	Range	Applications	REF.
Phase-shifted modal interferometers	1–1050 Hz	Various vibration measurements	[16]
MI* composed of seven-core fan-in coupler	1 kHz–12 kHz	Long-distance vibration measurement	[17]
Seven-core fiber SMI	2–2500 Hz	A variety of practical applications	[18]
Phase-shifted modal interferometers	1 mHz–30 Hz/0.76 mg–29.64 mg (acceleration)	Accelerometers	[20]
Dual-core fiber MI	Below 500 Hz	Accelerometers	[42]
Dual-core fiber MZI*/Seven-core fiber MZI	Heartbeat rate (69–100 bpm)/respiration rate (9–12 bpm)	Contactless vital signs monitoring for healthcare	[43]
Two counter-propagating MZIs in a 38.5 km seven-core fiber	― *	Long-distance distributed vibration location sensing	[45]
Four-core fiber MZI	Only demonstrated 100 Hz vibration sensing	Optical vibration sensor	[70]
Three-core fiber MZI	1–20 Hz	Long-term monitoring of construction and mechanical structures	[71]
Seven-core fiber SMI *	40–760 Hz in experiment and 50 Hz–25 KHz in theory	Diverse vibration sensing applications	[113]
Packaged seven-core fiber SMI	30–240 Hz	Low-frequency vibration sensing	[114]
Seven-core fiber SMI	60–290 Hz	Commercial vibration sensing applications	[115]
Three-core fiber SMI	5–100 Hz	Low-frequency and low-amplitude seismic sensing	[116]
Seven-core fiber SMI	5–50 Hz/0.00 g–0.04 g (acceleration)	Low-frequency accelerometers	[117]

* “―” means “not mentioned”; Michelson interferometer (MI); Mach–Zehnder interferometer (MZI); supermode interferometer (SMI).

**Table 3 sensors-23-03436-t003:** Comparison of MCF interferometer-based temperature sensors.

Sensor Structure	Sensitivity (pm/°C)	Range (°C)	REF.
Seven-core fiber MI with spherical reflective structure	165	Up to 900	[25]
Tapered seven-core fiber MZI with thermo-optical materials covered	Up to 25 nm/°C	10–50	[26]
Fiber-ball-coupled seven-core fiber MI *	70.6	20–90	[37]
Seven-core fiber MZI *	59.02	22–55	[57]
Lateral offset-fused seven-core few-mode fiber MZI	223.6	20–60	[76]
Tapered seven-core fiber MZI	36.8	Up to 1000	[87]
Tapered seven-core fiber MZI	89.19	24–130	[88]
Seven-core fiber MZI coupled with fiber ball	76.38	50–130	[89]
Lateral offset-fused seven-core fiber MZI	130.6	25–85	[90]
Seven-core fiber SMI *	48	Up to 1000	[97]
Seven-core fiber SMI	52	Up to 1000	[98]
Reflective seven-core fiber SMI	35	72–351	[99]
Three-core fiber SMI	29.426	−25–900	[100]

* Michelson interferometer (MI); Mach–Zehnder interferometer (MZI); supermode interferometer (SMI).

**Table 4 sensors-23-03436-t004:** Comparison of the MCF interferometer-based refractive index sensors.

Sensor Structure	Sensitivity (nm/RIU)	Range (RIU)	REF.
Three-core fiber MI *	−151.56 dB/RIU	1.3335–1.3720	[6]
Cladding etched asymmetrical dual-core fiber MI	270	1.34–1.38	[10]
Parallel dual FPI *	1096.9	About 1.34–1.37	[14]
Four-core fiber MZI *	113.27	1.3353–1.3549	[58]
Cladding etched asymmetrical dual-core fiber MZI	3119	1.3160–1.3943	[29]
Cladding etched seven-core fiber MZI	−1802.26	1.427–1.442	[77]
Exposed-core seven-core fiber MZI	−287.252 dB/RIU	1.3364–1.3679	[79]
Tapered seven-core fiber MZI	35089.28	1.4430–1.4442	[80]
Tapered seven-core fiber MZI	1280.94 dB/RIU	About 1.0005–1.0030	[81]
Tapered seven-core fiber MZI with graphene-coating	9194.6	1.4264–1.4278	[82]
Tapered micro-three-core fiber MZI	5815.50	About 1.332–1.341	[83]
Tapered four-core fiber MZI	171.2	1.3448–1.3774	[84]
Tapered seven-core fiber MZI	1435.76	1.3330–1.3451	[85]
Cladding etched seven-core fiber MZI	On the order of 10^4^	About 1.448–1.4476	[105]

* Fabry–Perot interferometer (FPI); Michelson interferometer (MI); Mach–Zehnder interferometer (MZI).

## Data Availability

Not applicable.

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
