# Peer review of "Advances in Multicore Fiber Interferometric Sensors"

_sensors, 2023, doi:10.3390/s23073436_

Round 1
Reviewer 1 Report
Multicore fiber interferometric sensors have an important research and application value in various physical and chemical parameters such as temperature, strain, curvature, refractive index, vibration, flow, torsion, etc. In this paper, different types of MCF interferometric sensors and recent developments are comprehensively reviewed in detail. The basic configurations and operating principles are introduced for each interferometric structure, and the performances of various MCF interferometric sensors for different applications are compared. In my point of view, the framework of this manuscript is clear and various MCF interferometric sensors for different applications are introduced basically completely. However, some important details need to be added and further explained. As a result, I recommend the manuscript could be published on sensors after some minor revisions.
Detailed comments and suggestions are described as follows:
1. I suggest the authors give the reflectance spectrum of the MCF-based Michelson interferometer. Otherwise, it might be difficult to understand “optimize the interference spectrum quality of MMI” and “a high fringe visibility of 25 dB has been achieved”.
2. The authors mentioned that by using the multiple channels in the MCF, multiple-path interference-based MMZI can achieve multi-parameter measurement (line 338). But they didn’t provide any measuring mechanism or reported works here. I think this issue should be more broadly discussed.
3. The authors discussed four different multicore fiber interferometric sensors in detail. Yet a short comparison of the advantages and disadvantages of four types of sensors should be added to make the discussions more intuitive.
4. The unit of refractive index sensitivity in line 9 of Table 4 should be dB/RIU instead of dB/RI.
5. The authors only compared the frequency bandwidth of the MCF interferometer-based vibration sensors. This is a significant drawback since it is impossible to understand the characteristic of vibration sensors without vibration/acceleration sensitivity. I think the author should add more details on this issue.
6. In addition, some details need to be modified, for example, the abbreviations for authors’ names in the reference should be unified. Besides, some figures look quite fuzzy, I suggest authors beautify them.
Author Response
Response to Reviewer 1 Comments
We would like to express our sincere thanks to the reviewer for his/her valuable comments and suggestions. We have revised the manuscript in accordance with the reviewer’s comments and suggestions. All the changes made in the revision are highlighted in red and underlined. Our replies to the reviewer’s comments and suggestions are as follows.
Point 1: I suggest the authors give the reflectance spectrum of the MCF-based Michelson interferometer. Otherwise, it might be difficult to understand “optimize the interference spectrum quality of MMI” and “a high fringe visibility of 25 dB has been achieved”.
Response 1: Thanks for your helpful comments. In the revised manuscript, we have added the reflectance spectrum of the introduced MCF-based Michelson interferometer with spherical end in Figure. 8 (b), line 225, additionally we have also added a comment in line 221-224, which points out more than 10 dB improvement in terms of the fringe visibility than the normal MCF-based Michelson interferometers.
Point 2: The authors mentioned that by using the multiple channels in the MCF, multiple-path interference-based MMZI can achieve multi-parameter measurement (line 338). But they didn’t provide any measuring mechanism or reported works here. I think this issue should be more broadly discussed.
Response 2: Thanks for your meaningful comments. In the revised manuscript, we have explained the work mechanism and listed some reported works on multi-path interference-based MZI in line 420-433.
Point 3: The authors discussed four different multicore fiber interferometric sensors in detail. Yet a short comparison of the advantages and disadvantages of four types of sensors should be added to make the discussions more intuitive.
Response 3: Thanks for your helpful comments. In order to make the discussion intuitive, we have compared the advantages and disadvantages of the four types of sensors in the revised manuscript in section 3 line 596-618.
Point 4: The unit of refractive index sensitivity in line 9 of Table 4 should be dB/RIU instead of dB/RI.
Response 4: Thanks for your attentive comments. We have corrected the mistake in the revised manuscript.
Point 5: The authors only compared the frequency bandwidth of the MCF interferometer-based vibration sensors. This is a significant drawback since it is impossible to understand the characteristic of vibration sensors without vibration/acceleration sensitivity. I think the author should add more details on this issue.
Response 5: Thanks for your insightful comments. The vibration/acceleration sensitivity is a crucial parameter to evaluate the performance of vibration sensors and accelerators, which indicates the response amplitude under different vibration intensities. We have checked the relevant articles [28,29,31,33,55,56,105-112] However it turns out that most of the publications didn’t give the parameter of vibration/acceleration sensitivity, apart from [29] and [112]. Besides, most of these publications clarified that their sensors have the advantage of high sensitivity [28,31,33,55,56,105,106,109-112], because the MCF interferometers are sensitive to curvature change which will contribute to high sensitivity of the vibration measurement. According to your suggestion, we have made modifications in Table 2 and added a discussion in line 656-660.
Point 6: In addition, some details need to be modified, for example, the abbreviations for authors’ names in the reference should be unified. Besides, some figures look quite fuzzy, I suggest authors beautify them.
Response 6: Thanks for your helpful comments. In the revised manuscript, we have added the abbreviations for authors’ names in the reference and beautified the figures to make them clearer (Figure 3-5, 7, 8-18, 20,22).
Reviewer 2 Report
In this paper, authors presented a review of multicore fiber interferometric sensors. Due to the specificity of fiber structure, i.e., multiple cores integrated in only one fiber cladding, multicore fiber interferometric sensors exhibit many desirable characteristics compared with the traditional fiber interferometric sensors based on single core fibers, such as structural and functional diversity, high integration, and space-division multiplexing capacity, etc. Because of some unique advantages over the standard fiber such as simple fabrication, compact size, and good robustness, MCF interferometric sensors have been developed to measure various physical and chemical parameters such as temperature, strain, curvature, refractive index, vibration, flow, torsion., etc. The basic configurations, operating principles are introduced for each interferometric structure, and eventually the performances of various multicore fiber based interferometric sensors for different applications are compared, including curvature sensing, vibration sensing, temperature sensing and refractive index sensing. Overall, this paper present a complete review and recent developments of the multicore fiber specifically used for the different sensing parameters. This article recommended for the publication.
Author Response
Response to Reviewer 2 Comments
Comments:In this paper, authors presented a review of multicore fiber interferometric sensors. Due to the specificity of fiber structure, i.e., multiple cores integrated in only one fiber cladding, multicore fiber interferometric sensors exhibit many desirable characteristics compared with the traditional fiber interferometric sensors based on single core fibers, such as structural and functional diversity, high integration, and space-division multiplexing capacity, etc. Because of some unique advantages over the standard fiber such as simple fabrication, compact size, and good robustness, MCF interferometric sensors have been developed to measure various physical and chemical parameters such as temperature, strain, curvature, refractive index, vibration, flow, torsion., etc. The basic configurations, operating principles are introduced for each interferometric structure, and eventually the performances of various multicore fiber based interferometric sensors for different applications are compared, including curvature sensing, vibration sensing, temperature sensing and refractive index sensing. Overall, this paper presents a complete review and recent developments of the multicore fiber specifically used for the different sensing parameters. This article recommended for the publication.
Response:We would like to express our sincere thanks to the reviewer for his/her positive comments. As the rapid development of multicore fiber, it shows great potential in sensing applications. Thanks to the structural specificity of multicore fiber, i.e., multiple spatial optical channels in only one fiber cladding, multicore fibers can provide an excellent platform to fabricate diverse functional lab-on-fiber devices, including multicore fiber gratings, multicore fiber interferometers and many other spatially integrated hybrid devices, etc. In this paper, we have introduced the basic configurations, operating principles for each interferometric structure, and compared the performances of various multicore fiber based interferometric sensors for different applications, including curvature sensing, vibration sensing, temperature sensing, refractive index sensing, etc. In conclusion, compared with traditional single core fiber-based interferometers, multicore fiber interferometers exhibit some desirable advantages such as structural and functional diversity, high integration, and space-division multiplexing capacity, etc., among which the exciting vector bending measurement ability of multicore fiber interferometers shows great potential in 3-D shape sensing, and long-term structure monitoring. We are very pleased to introduce these latest development and related work of MCF interferometric sensors. Thank you again for your encouraging comment!
Reviewer 3 Report
The manuscript "Advances in Multicore Fiber Interferometric Sensors" by Zhiyong Zhao et al is devoted to a review of multicore fiber interferometric sensors. This paper comprehensively reports on the recent investigations on MCF interferometric sensors, including the configurations, operating principles and applications etc. of the different types of interference sensors. The work is written very clearly and very honestly. The introduction is fully relevant to the performed studies, and the references are properly cited. In my opinion, this work provides systematical review about the research progress of interferometers in recent years, and surely merits the publication in Sensors after the authors incorporates the following points.
1) The section one adds the design and manufacturing process of multicore fiber. The structure design and their arrangement of multicore fiber includes two cores, three cores, seven cores fiber etc., and the manufacturing process of multicore fiber, including fiber prefabrication rod technology and drawing technology etc.
2) Ref. 2, 7, 16,46,55, 68, 85, 88,93, 102, 106,108 needs to be corrected (currently incomplete)
3) The resolution of some of the figures as submitted may not be high enough for production, such as Figure 6. In Figure 21, the figure captions of “a”, “b” and “c” should be changed as “(a)”, “(b)” and “(c)”. Please current it. The inset of Figure 22(b) is too small to recognize the digits.
Author Response
Response to Reviewer 3 Comments
We would like to express our sincere thanks to the reviewer for his/her valuable comments and suggestions. We have revised the manuscript in accordance with the reviewer’s comments and suggestions. All the changes made in the revision are highlighted in red and underlined. Our replies to the reviewer’s comments and suggestions are as follows.
Point 1: The section one adds the design and manufacturing process of multicore fiber. The structure design and their arrangement of multicore fiber includes two cores, three cores, seven cores fiber etc., and the manufacturing process of multicore fiber, including fiber prefabrication rod technology and drawing technology etc.
Response 1: Thanks for your insightful comments. The introduction of the design and manufacturing process of multi-core optical fiber is helpful for readers to have a better understanding of multicore fiber structure and the parameters that will impact the sensing performance. In the revised manuscript, we have briefly introduced the manufacturing process of multi-core fiber in line 37-39. Besides, the design of multi-core optical fiber involves a lot of contents, including core number, core diameter, material, core-to-core distance, and group dispersion, group delay, transmission loss, etc. For multicore fibre interferometric sensors, the most important parameter is the crosstalk that will determine the interference type. In addition, the core diameter, core-to-core distance, material and cladding thickness will also impact the sensing performance such as sensitivity. In the previous manuscript, we have introduced the core-to-core distance and crosstalk of multicore fiber in line 53-65, Figure 2. In the revised manuscript, we add a discussion about the impact of sensing performance brought by core diameter, material and cladding thickness in line 65-70.
Point 2: Ref. 2, 7, 16,46,55, 68, 85, 88,93, 102, 106,108 needs to be corrected (currently incomplete)
Response 2: Thanks for your attentive comments. In the revised manuscript, we have corrected the format of these references.
Point 3: The resolution of some of the figures as submitted may not be high enough for production, such as Figure 6. In Figure 21, the figure captions of “a”, “b” and “c” should be changed as “(a)”, “(b)” and “(c)”. Please current it. The inset of Figure 22(b) is too small to recognize the digits.
Response 3: Thanks for your helpful comments. In the revised manuscript, we have beautified these figures to make them clearer (Figure 3-5, 7, 8-18, 20,21,22) and the figure captions in Figure 21 have been corrected.
Reviewer 4 Report
This review entitled “advances in multicore fiber interferometric sensors” falls within the scope of the Sensors. The presented review is interesting and well written, and therefore, can be considered for publication in the Sensors. However, some revisions are required before it can be considered for publication as follows:
1. A sub-section is needed to explain the essential parameters of multicore fiber interferometric sensors such as quality factor, sensitivity, detection limit, and so on.
2. Adding a Table to list all symbols and abbreviations can be useful for readers of this review paper.
3. Is there any polarization consideration effect on the sensor performance? Please clarify.
4. There are two references for multicore photonic crystal fibers in the present review; [52] and [62]. It is recommended to add some related works such as: 10.1007/s10825-022-01933-6; 10.3390/s19030464; 10.1109/ICSENS.2010.5691019.
5. The quality of some figures is low, for example, Figure 17b.
Author Response
Response to Reviewer 4 Comments
We would like to express our sincere thanks to the reviewer for his/her valuable comments and suggestions. We have revised the manuscript in accordance with the reviewer’s comments and suggestions. All the changes made in the revision are highlighted in red and underlined. Our replies to the reviewer’s comments and suggestions are as follows.
Point 1: A sub-section is needed to explain the essential parameters of multicore fiber interferometric sensors such as quality factor, sensitivity, detection limit, and so on.
Response 1: Thanks for your insightful comments. Essential parameters like quality factor, sensitivity, detection limit, are critical parameters to evaluate the performance of multicore fiber interferometric sensors. We have checked all the references about these aspects. To make the discussions more intuitive. we have compared each structure of multicore fiber interferometric sensors about their advantages and disadvantages to indicate the detection limits of each structure (section 3 line 596-620). On the other hand, the sensitivity enhancement, directional dependence of sensitivity of the vector bending sensor, and the sensitivity comparison of several specific application fields, including curvature, vibration, temperature, and refractive index sensing have also been discussed in the manuscript. As for the quality factor, since the reported MCF interferometric sensors were developed for various applications, the parameters mentioned in each publication is quite different and incomplete, it is difficult for us to collect full information for a quality factor, so we are very sorry that it is difficult for us to establish a quality factor in this case.
Point 2: Adding a Table to list all symbols and abbreviations can be useful for readers of this review paper.
Response 2: Thanks for your attentive comments. According to your suggestion, we have added a table, which lists all the abbreviations used in this paper in Table 5.
Point 3: Is there any polarization consideration effect on the sensor performance? Please clarify.
Response 3: Thanks for your insightful comments. We have checked all the references about the issue of polarization effect on the MCF interferometers and the sensor performance. However, it turns out that this issue is seldomly discussed in those publications. As a matter of fact, according to the principle of beam interference, the polarization of each beam will impact the quality of output interference signal, i.e., the fringe visibility of optical interference spectrum. Therefore, in order to obtain good interference spectrum with high fringe visibility, the polarization state of interference light needs to be adjusted carefully in the experiment. We have added a discussion in the revised manuscript in line 618-620.
Point 4: There are two references for multicore photonic crystal fibers in the present review; [52] and [62]. It is recommended to add some related works such as: 10.1155/2012/598178; 10.1007/s10825-022-01933-6; 10.3390/s19030464; 10.1109/ICSENS.2010.5691019.
Response 4: Thanks for your comments. Photonics crystal fibers have many desirable characteristics for sensing applications, e.g., sensitive to vibration, bending, and hydraulic pressure, etc. We have added some more references in the revised manuscript, i.e., 10.3390/s19030464 [115] and 10.1155/2012/598178 [116]. Besides, the advantages of multicore photonic crystal fibers have also been introduced in section 2 line 353-356.
Point 5: The quality of some figures is low, for example, Figure 17 b.
Response 5: Thanks for your helpful comments. In the revised manuscript, we have beautified most of the figures to make them clearer (Figure 3-5, 7, 8-18, 20,21,22) and the figure captions in Figure 21 have also been corrected.
Round 2
Reviewer 4 Report
I would like to thank the authors for improvement of the manuscript considering the comments. The comments 1 and 4 need to be reconsidered in the final revision. I recommend to give more details of comment 1 in the manuscript and use the recommended references of comment 4. In addition the references should be cited in order in the text (in the present form [1], [2], [3], [4], [5], [6], [7], [20], [24], [42], and so on).
Author Response
Response to Reviewer 4 Comments
We would like to express our sincere thanks to the reviewer for his/her valuable comments and suggestions. We have revised the manuscript in accordance with the reviewer’s comments and suggestions. All the changes made in the revision are highlighted in red and underlined. Our replies to the reviewer’s comments and suggestions are as follows.
Point 1: I recommend to give more details of comment 1 in the manuscript. (A sub-section is needed to explain the essential parameters of multicore fiber interferometric sensors such as quality factor, sensitivity, detection limit, and so on.)
Response 1: Thanks for your helpful comments. We have discussed the important parameters of multicore fiber interferometric sensors like sensitivity and limitations, which are critical to evaluate the performance of multicore fiber interferometric sensors, and these discussions are distributed in various parts of the article according to the presented sensing applications. To make the discussions more intuitive, we have compared each structure of multicore fiber interferometric sensors about their advantages and disadvantages to indicate the detection limits of each structure (section 3 line 596-620). In addition, the limitations of multicore fiber Mach-Zehnder interferometers and Michelson interferometers for bending sensing have been discussed in section 3 line 641-646. On the other hand, the sensitivity enhancement of temperature/refractive index sensor (section 2 line 320-329, 446-465, and section 3 line 672-678, 684-696), directional dependence of sensitivity of the vector bending sensor (line 136-142, 258-272, 278-288, 367-386, 531-540), and the sensitivity comparison of several specific application fields, including curvature (table 1 and line 632-641), vibration (table 2 and line 654-660), temperature (table 3 and line 672-678), and refractive index (table 4 and line 684-696) sensing have also been discussed in the manuscript. As for the quality factor, we have checked all the references to give a discussion. However, since the reported MCF interferometric sensors were developed for various applications, the parameters mentioned in each publication is quite different and incomplete, it is difficult for us to collect full information for a quality factor, so we are very sorry that it is difficult for us to establish a quality factor in this case.
Point 2: I recommend to use the recommended references of comment 4. (There are two references for multicore photonic crystal fibers in the present review; [52] and [62]. It is recommended to add some related works such as: 10.1155/2012/598178; 10.1007/s10825-022-01933-6; 10.3390/s19030464; 10.1109/ICSENS.2010.5691019.)
Response 2: Thanks for your comments. We have added all the recommended references in the revised manuscript in line 353-356 ([37-40]).
Point 3: In addition the references should be cited in order in the text (in the present form [1], [2], [3], [4], [5], [6], [7], [20], [24], [42], and so on).
Response 3: Thanks for your attentive comments. We have rearranged all the references after adding the recommended articles. Note that the references in line 82-112 are not cited in order, because these references are mentioned more than once and are cited in order in the following article. In this way, the order of articles in the same type can be very close, and the citation numbers can be very concise when a large number of articles are cited in the following text. For example, [11-16] in line 134, [69-79] in line 441, and [105-112] in line 552. Otherwise, the following citation numbers will be very scattered and look very lengthy, which is not convenient for readers to quickly find articles in the same type.
Round 3
Reviewer 4 Report
The manuscript is again studied and the revisions are considered. The authors have addressed the raised comments, and the present version of the manuscript satisfies my comments. Accordingly, I recommend it for publication in Sensors.